# Indicators of Climate Change, Geospatial and Analytical Mapping of Trends in India, Pakistan and Bangladesh: An Observational Study

**DOI:** 10.3390/ijerph192417039

**Published:** 2022-12-19

**Authors:** Faiqa Falak, Farsom Ayub, Zunaira Zahid, Zouina Sarfraz, Azza Sarfraz, Karla Robles-Velasco, Ivan Cherrez-Ojeda

**Affiliations:** 1Department of Research, Services Institute of Medical Sciences, Lahore 54000, Pakistan; 2Department of Research and Publications, Fatima Jinnah Medical University, Lahore 54000, Pakistan; 3Department of Pediatrics and Child Health, The Aga Khan University, Karachi 74800, Pakistan; 4Department of Allergy, Immunology & Pulmonary Medicine, Universidad Espíritu Santo, Samborondón 092301, Ecuador

**Keywords:** air quality, analytical mapping, Bangladesh, climate change, drought, flooding, geospatial, India, Pakistan, South Asia

## Abstract

The year 2022 has served as a recall for the impact that climate change has in the South Asian region, which is one of the most vulnerable regions to climate shock. With a paucity of climate-based and geospatial observational studies in South Asia, this paper (i) links power sectors and carbon dioxide emissions, (ii) maps nitrogen dioxide density across three countries (Pakistan, India, and Bangladesh), (iii) understands electricity generation trends and projects weather changes through 2100. We monitored data monitored between 1995 and 2021. The following databases were used: the International Energy Agency, the World Bank, the UN Food and Agricultural Organization. Raw data was obtained for climate indicators, which were entered into Microsoft Excel. Geospatial trends were generated in the ArcGIS geostatistical tool by adopting the ordinary kriging method to interpolate and create continuous surfaces depicting the concentration of nitrogen dioxide in the three countries. We found increased usage of coal and fossil fuels in three countries (Pakistan, India, and Bangladesh). Both were significant contributors to carbon dioxide emissions. The geographic localities in South Asia were densely clouded with nitrogen dioxide as reported with the tropospheric column mapping. There are expected to be increased days with a heat index >35 °C, and consecutive dry days from 2020 and 2100. We also found increased chances of flooding in certain regions across the three countries. This study monitored climate change indicators and projects between 1995 and 2100. Lastly, we make recommendations to improve the relationship of the environment and living beings.

## 1. Introduction

The year 2022 has served as a recall of the World Bank’s prior warnings about climate change in the South Asian region since the onset of the 21st century [1,2,3,4,5]. The floodings in Pakistan have led to a humanitarian crisis where nearly 2000 residents faced death and 33 million (15% of the population) were homeless [6,7]. In Pakistan, nearly $40 billion worth of damages were noticed and four million hectares (ha) of diverse agricultural land were lost [6,7]. On the other hand, regions in India including Uttar Pradesh and West Bengal witnessed severe rainfall deficits, whilst flooding occurred in Andhra Pradesh, Madhya Pradesh, and Maharashtra [8,9]. Typically, the four months of monsoon that South Asia witnesses provide 80% of the annual rainfall. However, with climate change accelerating the water cycle, certain provinces and cities have seen 400–500% more rainfall than the anticipated yearly average [10,11]. Furthermore, air pollution has spanned New Delhi, Lahore, and Dhaka, leading to severe air disasters impacting the general health of the residents [12,13]. Notably, reports find that G20 countries are collectively responsible for 80% of global emissions, whereas Lahore (Pakistan), New Delhi (India), and Dhaka (Bangladesh)—three South Asian countries hosting 1.8 billion residents—are facing large adverse effects of climate change [14,15,16].

The World Meteorological Organization (WMO), a UN agency, has found that if greenhouse emissions remain high, the global temperature is expected to rise by three degrees in the second half of the 21st century [17,18]. Across densely populated areas in South Asia including North India, Bangladesh, and Pakistan, the temperature is expected to rise by at least 20% [19,20,21]. The WMO report outlines that the ozone level increase in the troposphere is a direct consequence of emissions from fossil fuel combustion, whereas 20% of the increase is due to climate change. Causes include deforestation, agriculture/other land use changes, and energy-related carbon-dioxide emissions from buildings [22]. The cycle of air pollution upsurges the risk of a ‘climate penalty’ in the region beyond its borders [23].

South Asia is one of the most vulnerable regions to climate shocks [24]. Both Bangladesh and Pakistan included four large provinces [25]. Deforestation and water pollution have been major environmental concerns with recent air pollution trends being a central concern [26]. Pakistan and India have the same geographical location and are located along the tectonic plate boundary [27]. On the other hand, Bangladesh is a low-lying country with the Meghna River, Ganges River, and Brahmaputra River flowing in the Bay of Bengal [28,29]. Flooding has long been a concern for Bangladeshi residents that occurs regularly in the country [30]. India consumes considerable coal for electricity production, which contributes to air pollution [31,32]. Deforestation is also a common occurrence across rural areas and water pollution, particularly along the Ganges River, is severe and affects a large proportion of Indian residents [25]. 

The current literature cites reports of geospatial and analytical studies conducted in the region. Syed and colleagues (2021) assessed changes in season trends and spatial patterns of temperature indicators between 1979 and 2014 [33]. The authors used datasets by Climate Forecast System Reanalysis (CFSR). The authors found significant trends in autumn and spring of minimum temperature in Punjab, Pakistan [33]. Overall, the authors reported the reasons of sustainability imbalance in Punjab, Pakistan. These included fluctuations in humidity and trends of high temperature [33]. Javid et al. (2019) used the remote sensing climate compound index (RSCCI) to measure humidity, aridity, and semi-aridity in Pakistan [34]. The authors found an increase in humid regions and wetlands in Pakistan at 9.7% and 1.9%, respectively [34]. There was a reduction in intense drought (4.2%) in the country between 2000 and 2018 [34]. The study concluded with predicting that the next 30 years may lead to dramatic climatic changes in Pakistan [34]. Ali et al. (2021) evaluated spatio-temporal rainfall patterns and trends in Pakistan between 1961 and 2020 by including 82 rainfall stations [35]. The authors used the Bayesian kriging regression prediction (EBKRP) technique [35]. The lowest rainfalls were predicted in Baluchistan. There was a significant downward trend of temporal rainfall evaluation in Sindh [35]. The study sheds light on environmental planning in the region [35].

Geospatial mapping and climate-based observational studies on South Asia have been published in the literature [12,36,37,38]; however, there is a paucity of data-based reports in this region. South Asia has large variability in geographical features, climate and landscapes [39]. With the rapid population growth and economic development, the region is falling behind in both building knowledge databases and sequentially analyzing existing data on climate indicators, land use and cover, water disasters and other geohazards [40]. 

The motivation behind this study is to prioritize climate adaptation and subsequently adapt strategies. There is an urgent need for quantitative-regional based assessment of climate risks. The aim of this observational study is to understand the linkage between power sectors in the region (India, Pakistan, and Bangladesh) and carbon dioxide emissions, map areas with nitrogen dioxide tropospheric columns and correlated air quality indices, understand the power sector and all energy generation trends in the region, and lastly to project weather changes throughout the 21st century. 

## 2. Materials and Methods

The data utilized for this study was monitored between the years (1995–2021) and the trends estimates were made until the year 2100 among three South Asian countries (Pakistan, India, and Bangladesh) with world estimates used for comparative purposes. This observational study obtained data from international and national data banks in addition to governmental and non-governmental websites. These included: Ember Climate Open-Source Data (https://ember-climate.org/data/data-explorer/, accessed on 11 November 2022), and datasets by the UN Food and Agriculture Organization (FAO). (https://fra-data.fao.org/, accessed on 11 November 2022) for deforestation and agricultural data sourced from the UN FAO Forest Resources Assessment. The International Energy Agency (IEA) database (https://www.iea.org/, accessed on 11 November 2022) was utilized for power sector capacities and the growth of renewable energy generation by region and technology. The World Bank’s Climate Change Knowledge Portal (https://climateknowledgeportal.worldbank.org/, accessed on 11 November 2022) was utilized for country-specific drought, dry days, and consecutive hot days outcomes.

We collected raw and ready-to-analyze data for all three countries (Pakistan, India, and Bangladesh) for climate indicators and stipulated trends with the latest trends until 2021 and no backend to the inception of data. Air pollution was reported in terms of CO_2_ emissions and nitrogen dioxide tropospheric levels with AQI indicators. Furthermore, the current state of climate control was measured with geospatial and analytical mapping of deforestation, agricultural land per capita, and ‘hot spots’ of nitrogen dioxide in the region. The Geographic Information System (GIS) tool was used to generate spatially distributed air pollution maps in India, Pakistan, and Bangladesh. In order to map the pollution levels using nitrogen dioxide, the ArcGIS geostatistical analysis tool was used due to its ability to handle wide data formats and layers in digital maps using frameworks of spatio-temporal analysis. The ordinary kriging method was adopted to interpolate and create continuous surfaces from the database of measured concentration across the locations [41,42]. Further analysis was conducted to ascertain the usage of different energy types in the power sector and reliance on fossil and coal as modes of production. All data was entered into Microsoft excel and output plots were generated to depict the sequential trends in the various indicators. 

Search terms combined with a Boolean methodology were comprised of “South Asia”, “India”, “Pakistan”, “Bangladesh”, “Power”, “Emissions”, “Carbon Dioxide”, “Air Pollution”, “Nitrogen Dioxide”, “Agricultural”, and “Deforestation”. Data reports and repositories that addressed these areas were reviewed. The following is a list of key terminologies and verbiage used throughout this study:i.**Agricultural area per capita:** Agricultural land is the total of land and cropland used as pasture for grazing livestock;ii.**Deforestation:** The permanent conversion of natural forest land to other uses including infrastructure development, settlements, and shifting cultivation;iii.**AQI:** The air quality index (AQI) scale measures the number of pollutants in a specific region at a selected time interval—to help assess the quality of air;iv.**CO_2_:** Carbon dioxide is a colorless greenhouse gas formed when any material containing carbon is combusted;v.**Coal:** A brownish-black or black sedimentary rock that is typically burned for fuel to generate electricity;vi.**Dry days:** Days with rainfall <0.85 mm within a typically defined period of 24 h;vii.**Fossil fuel:** A class of hydrocarbon-containing materials of biological origin that can be used as a source of energy;viii.**Heat index:** An equation that combines relative humidity and air temperature to determine the perceived temperature;ix.**kWh:** A unit of energy, which means one kilowatt hour (kWh) of power;x.**Mean drought index** (projected change, unitless): The standardized precipitation evapotranspiration index (SPEI), or mean drought index, calculated for a specific period, is closely related to drought impacts on ecosystems, water, and crop resources. The SPEI accounts for potential evapotranspiration and precipitation in determining drought;xi.**Nitrogen dioxide tropospheric column:** A two-dimensional field of the total amount of nitrogen dioxide molecules per unit area in an atmospheric column extending from the Earth’s surface to the tropopause;xii.**NO_2_:** Nitrogen dioxide is a gaseous air pollutant composed of nitrogen and oxygen;xiii.**NO_x_:** Gases nitric oxide and nitrogen dioxide that is produced when fuel is burned;xiv.**Power sector emissions:** Emissions of gas or radiation into the atmosphere due to power sectors.

This study was exempt from Ethical Review Board committee approval given the secondary nature of the datasets and analysis.

## 3. Results

In this paper, we studied the indicators of climate change that ascertained both causation and effect. The first key aspect comprised the power sector’s use of coal and fossil along with emissions of carbon dioxide. The second central finding related to the geographic mapping of areas with nitrogen dioxide tropospheric column mapping and indicators of the AQI; this was further supported by annual deforestation and agricultural area per capita in the region. Thirdly, power sector and electricity generation trends in India, Southeast Asia, and the world were extrapolated. Lastly, projected weather changes in the heat index (days surpassing 35 °C) and consecutive dry days until 2039, and the annual SPEI drought index for the years 2020–2100 were presented.

### 3.1. Power Sector Usage and Emissions of Coal, Fossil, and Carbon Dioxide

#### 3.1.1. Power Section Emissions Due to Coal and Fossil (2018–2021)

Between 2018 and 2021, the power sector emissions due to coal first decreased from 2.62 (2018) to 2.56 (2019) megatons of CO_2_ but increased to 8.18 megatons of CO_2_ by 2021 in Bangladesh [43]. In India, the emissions due to coal use decreased between 2018 and 2020 from 983.07 (2018) to 923.13 megatons of CO_2_ and increased marginally to 1042.33 megatons of CO_2_ [43]. In Pakistan, the emissions overall increased from 6.99 (2018) to 11.42 (2021) megatons of CO_2_, but there was a slight shift from 11.24 (2019) to 10.19 (2020) megatons of CO_2_ [43] (Figure 1).

In Bangladesh, the power sector emissions due to fossil use were 39.89 megatons of CO_2_ in 2018 and increased to 47.77 megatons of CO_2_ in 2021 [43]. In India, however, the emissions decreased from 1022.33 (in 2018) to 961.1 (in 2021) megatons of CO_2_, with a final increase to 1075.4 megatons of CO_2_ in 2021 [43]. In Pakistan, there was an overall upshift of emissions from 38.09 megatons of CO_2_ in 2018 to 43.41 megatons of CO_2_ in 2021 [43] (Figure 1).

#### 3.1.2. Emissions Intensity of Carbon Dioxide per kWh (2018–2021)

The emissions intensity of CO_2_ per kWh increased from 539.876 kWh (in 2018) to 570.985 kWh (in 2021) in Bangladesh [43]. In India, on the other hand, there was a decline from 655.366 kWh in 2018 to 637.182 kWh in 2021 [43]. Pakistan saw a steady rise from 334.351 kWh in 2018 to 358.056 kWh in 2020 and an increase to 347.26 kWh in 2021 [43] (Figure 2).

#### 3.1.3. Electricity Generated Using Coal and Fossil in Terawatt Hours (2018–2021)

The electricity generated using coal was 3.2 Terawatt hours in 2018 and 9.98 Terawatt hours in 2021 in Bangladesh [43]. On the other hand, in India, there was a rise from 1198.86 Terawatt hours in 2018 to 1271.14 Terawatt hours in 2021 [43]. Pakistan witnessed a steady rise from 8.53 Terawatt hours to 13.93 Terawatt hours in 2021 [43] (Figure 3).

On assessing the energy generated using fossil, Bangladesh generated 72.57 Terawatt hours in 2018 and 82.52 Terawatt hours in 2021 [43]. In India, there was a rise from 1276.32 Terawatt hours of production in 2018 to 1337.63 Terawatt hours in 2021, with a decrease in 2020 (1202.34 Terawatt hours) [43]. Similarly, Pakistan presented a rise from 66.29 to 71.81 Terawatt hours of energy generation between 2018 and 2021 with a decrease in 2020 (64.47 Terawatt hours) [43] (Figure 3).

### 3.2. Geographic Mapping of Low Air Quality, Annual Deforestation, and Agricultural Area

#### 3.2.1. Geospatial Map of Low Air Quality Hot Spots in South Asia

With the geospatial map generated on 8 November 2022, densely polluted air localities were mapped in South Asia (Figure 4). Data sources comprised the European Commission, European Space Agency, Esri, and the UKRI GCRF South Asian Nitrogen Hub, [44,45]. A key summary of trends obtained is as follows:Lahore: Nitrogen dioxide tropospheric column (140 μmol/m^2^); Air quality index: 193 (US AQI)New Delhi: Nitrogen dioxide tropospheric column (262 μmol/m^2^); Air quality index: 251 (US AQI)Dhaka: Nitrogen dioxide tropospheric column (212 μmol/m^2^); Air quality index: 140 (US AQI)

#### 3.2.2. Map Representing Annual Deforestation between 2015 and 2020 in South Asia

Deforestation does not indicate net forest loss but also accounts for any gains in the forest over the given time period. Since 2010, the net forest loss worldwide was 4.7 million ha per year, however, the deforestation rates were higher [46] (Figure 5). It is estimated that the cumulative deforestation was 10 million ha every year since 2010 [46]. Rated at six scale points (1 = 0–10,000 ha; 2 = 10,000–50,000 ha; 3 = 50,000–100,000 ha; 4 = 100,000–500,000 ha; 5 = 500,000–1 million ha; 6= >1 million ha), India was scaled 5 with, 668,400 ha lost per year [46]. While no data was indicated for Pakistan, Bangladesh was scaled at 2, with 18,190 ha lost per year [46].

#### 3.2.3. Map Depicting Agricultural Area per Capita in 2018 across South Asia

The worldwide peak of deforestation was witnessed in the 1980s and has declined ever since. Since 1961, the agricultural land has increased by 7% whereas the global population has increased by 147% during the same period, reaching 7.6 billion people; this means that the agricultural land per individual has more than halved from 0.63 ha to 1.45 ha [46]. 

The map is scaled in nine points (1 = 0–0.25 ha; 2 = 0.25–0.5 ha; 3 = 0.5–1 ha; 4 = 1–2.5 ha; 5 = 2.5–5 ha; 6 = 5–10 ha; 7 = 10–25 ha; 8 = 25–50 ha; 9= >50 ha) [46] (Figure 6). All three countries fell under category 1, with agricultural land per capita between 0–0.25 ha [46]. The highest of the three was Pakistan, with 0.17 ha per capita [46]. This was followed by India with 0.13 ha per capita [46]. Finally, Bangladesh had 0.06 ha per capita [46] (Figure 6).

### 3.3. Power Sector and Electricity Generation Trends in India and the World 

Across the world, the annual growth of wind renewable energy generation in 2019 was 149 TWh, with India generating 2 TWh and Southeast Asia generating the lowest at 1 TWh in the same year [47] (Figure 7). On reviewing solar PV trends, the world had a growth of 131 TWh, with India at 10 TWh and Southeast Asia at 4 TWh [47]. Hydro energy growth across the world was 102 TWh, with India at 23 TWh and Southeast Asia at 4 TWh [47]. While bioenergy was growing at 50 TWh in 2019, India contributed to negative 3 TWh and Southeast Asia generated 2 TWh [47]. The 10-year average in Southeast Asia was 18 TWh, which was not comparable to the world (331 TWh) [47] (Figure 7). 

The coal annual power capacity addition in India between 2010 and 2019 ranged between 3.8 gigawatts (GW) to 19.3 GW with 10.7 GW in 2010 and 2019 [48] (Figure 8). The annual addition of gas power capacity in India was the highest in 2014 with 2.6 GW; commencing from 1.2 GW in 2010 to -0.3 GW in 2019 [48]. The capacity added to power in India between 2010 and 2019 was the highest in 2015 with 2 GW generated, whereas the lowest generation was witnessed in 2019 at 0.2 GW [48]. Power capacity added with wind usage was overall high with 2.3 GW generated in 2010 and 2.4 GW in 2019; the highest documentation was for 2017 with 4.1 GW [48]. While solar PV power capacity addition was negligent in 2010, the generation rose to 9.3 GW in 2019 [48]. With a continual rise in wind and solar usage, the total capacity increased from 6.7 GW in 2010 to 18.2 GW in 2019 (an estimated 18% of the total power sector capacity) [48] (Figure 8).

### 3.4. Projected Changes in Weather Conditions from 2020–2100

#### 3.4.1. Heat Index Surpassing 35 °C Projections

The heat index measures apparent temperature which includes the influence of atmospheric moisture [49]. High temperature with high moisture leads to high heat indexes. With this, a mapping of the number of days where the heat index surpasses 35 °C during the data aggregation period [49]. The heat index also covers the seasonal heat risks and changing seasonal heat risks over time [49]. In this section, a collation of findings is presented for the years 2020–2029 [49].

In key regions across Pakistan, the number of days with a heat index >35 °C comprised Sindh (140.6 days), Punjab (108.33 days), and Baluchistan: 49.62 days (Figure 9). 

Across India, regions with the highest burden of the heat index (>35°C) included: West Bengal (136.49 days), Gujarat (128.55 days), Uttar Pradesh (120.47 days), Haryana (120.27 days), Rajasthan (119.93 days), and Punjab (110.73 days) (Figure 9).

Bangladesh had the highest stipulated increase in heat index (days per year exceeding 35 °C): Khulna (160.31 days), Rajshahi (159.08 days), Dhaka (155.63 days), and Rangpur (140.9 days) (Figure 9).

#### 3.4.2. Consecutive Dry Days Projections

Dry days are defined as any days where the daily accumulated precipitation is less than 1 mm. The indicator represents the maximum length of dry spells that are computed sequentially for the entire time series and takes into account the maximum value each month during the data period. Consecutive dry days aid us in understanding the changing precipitation patterns and the changing periods of aridity for the location. The identified national units reflect the trends for the projected period until 2039 [49] (Figure 10).

Key regions across Pakistan comprised the following with consecutive dry days: Sindh (340.53 days), Baluchistan (332.98 days), and Punjab (307.66 days).

In India, the regions with the highest estimates of consecutive dry days included: West Bengal (214.88 days), Gujarat (305.6 days), Uttar Pradesh (264.78 days), Haryana (287.54 days), Rajasthan (314.84 days), and Punjab (281.16 days).

Bangladesh represented consecutive dry days as well across the following regions: Khulna (218.33 days), Rajshahi (203.78 days), Dhaka (198.62 days), and Rangpur (193.37 days).

#### 3.4.3. Projected Annual SPEI Drought Index until the Year 2100

Drought can be expressed in multiple ways from simple precipitation deficits to complex estimates of remaining soil moisture. The SPEI is used as a global measure for drought monitoring over cumulative time intervals. Positive values are indicative of positive (or wet) water balance conditions, whereas negative values are indicative of negative (or dry) water balance. SPEI in this study is used to determine the onset, duration, and magnitude of drought conditions with respect to normal conditions of natural and managed systems including rivers, crops, ecosystems, and water resources [49].

Overall, Pakistan represented seemingly downward trends (negative) over the course of the estimated time period, meaning that the water balance was to be dry [49]. India represented overall static trends as compared to the reference period (1995–2014), meaning that while there were dry and water spells, overall, there were no large fluctuations in estimates [49]. On the other hand, Bangladesh showcased an upwards (positive) trend over the estimated time period where water spells were of note [49] (Figure 11).

## 4. Discussion

In this study, we reported indicators of four key areas of interest as described in Figure 12. These include (i) the power sector’s use of fossil and coal, and carbon dioxide emissions; (ii) nitrogen dioxide tropospheric column mapping and air quality; (iii) electricity generation sources and trends; (iv) projected weather changes for heat and extreme weather conditions.

A study by Rehman and colleagues (2022) analyzes the impact of electricity production from sources including coal, nuclear, natural gas, and oil and the impact on carbon dioxide emission using annual series data between 1975 and 2020 [50]. It is ascertained through a nonlinear autoregressive distributed lag technique, applied to study the influence of electrical energy on CO_2_ that emissions in Pakistan have faced negative shocks [50]. Another analytical study by Raza and colleagues (2022) employs the decoupling and decomposition methods in the driving factors of CO_2_ emissions due to electricity generation between 1990 and 2019 [51]. The findings suggest that the population and activity effect are key driving factors in the rising CO_2_ emissions; further, a correlation to activity, total electricity, electricity intensity, generation structure of electricity, the efficiency of energy, and the fuel emission factor effect were witnessed [51]. It has been considered essential to focus on all the aforementioned factors’ decoupling states to limit CO_2_ emissions [51]. However, nearly 84% of Pakistan’s energy demand relies on fossil fuel, which has contributed to rising CO_2_ emissions by 133.4 metric tons between 1990 and 2019 [52].

India is the second most populous country on earth and one of the most rapidly growing economies; the social and economic development has been largely subsidized due to ample and cheap fossil fuels [53]. The emission trends for pivotal greenhouse gases, including CO_2_ and other critical air pollutants, are expected to increase by 147% by 2025; this rapid increase is allocated to industrial development, which is followed by demand from agricultural and domestic sectors [54]. While the United States has historical significance in the consumption of coal, China and India together consumed 60% of the world’s primary coal in 2016 [55]. One reason is that coal power plants in India are at a considerably nascent age compared to high-income countries, thus allowing more useable life [55]. Together, India, China, and the United States account for 91% of the world’s coal production. The Indian subcontinent in particular is a large drive of coal and fossil demand with the idea of modernizing and industrializing the countries [55,56].

Bangladesh is an emerging economy, which largely generates electricity using coal. Electricity consumption increased by 526% between 2000 and 2021 [57,58,59], which was higher the other countries with similar economic development including Sri Lanka (79%) or Nigeria (94%) [57]. Based on historical trends, electricity consumption per capita may increase by 22 times by 2050, compared to in 2014 [60]. The country has been dominated by fossil fuel since 1971. However, despite the increased demand for electricity, the power generation sector cannot meet the demand of the population due to load shedding. Furthermore, Bangladesh is one of the top five countries for global coal and gas power projects in development as per the Global Energy Monitor [61]. In 2022, an estimated 20 GW of new, coal and gas-based power projects are under construction in southeastern Bangladesh in Chattogram [62]. Together, the power-based coal and fossil projects in the country have enabled carbon dioxide and other pollutants to rise in the region [63].

Studies find that among the numerous air pollutants, nitrogen oxides and pollutants such as lead and sulfur dioxide are major concerns in the Indian subcontinent [64,65]. As our study depicts in the GIS map, NO_2_ presence in the troposphere is a central factor that determines the quality of air and impacts human health and plant growth. Furthermore, the AQI utilized in our study ties in with photochemical smog that has detrimental effects associated with the high NO_2_ concentration [66,67]. An analysis determined that the many sources of NO_2_ emission in India comprise 32% transportation, 50% industrialization, and 10–20% accounted by the biomass burning sectors [68].

Comparable with the findings of reduction during the early 2020 period of CO_2_ emissions and improvement in air quality, Pakkattil et al. (2021) reported 82% reductions in volatile organic compounds across major metropolitan cities in India compared to the pre-lockdown period [69]. COVID-19-related restrictions were documented to have reduced particulate matter and trace gas concentrations worldwide. A temporal comparison of ozone monitoring instruments obtained NO_2_ column density during the lockdown period in India with counterfactual baseline concentrations of NO_2_ between 2015 and 2019 [70]. The NO_2_ concentration in urban areas of Delhi declined by 60% in phase 1 of the lockdown and was reduced by 3.4% during the fifth phase. However, no reductions were noted in rural areas. With NO_x_ emissions estimated using the TROPOspheric Monitoring Instrument (TROPOMI), urban Delhi and power plants in the region presented with an average decline of 72.2% and 53.4% [71]. A shortcoming of this technique is the lack of accounting for photochemical and dynamical conditions; these must be addressed by optimizing emission inventories.

The global impact of air pollution is estimated to have reached 7 million deaths per year due to exposure to household and outdoor air pollution. The common cause of death has been linked to lung cancer, as well as both acute respiratory diseases such as asthma and chronic pulmonary disease, heart disease, and stroke [72]. In 2019 alone, 1.67 million premature deaths occurred due to air pollution, whereas Pakistan had 236,000 deaths and Bangladesh saw 174,000 deaths. The World Health Organization has continued to emphasize the crucial the role of air pollution as a non-communicable disease, which is currently the second largest indicator following tobacco smoking. There are 4 billion people in the Asia and Pacific region, and 92% of the population that experience air pollutants at increased levels. The Organization for Economic Co-operation and Development (OECD) also reported that the gross domestic product across different economies in Asia will face a reduction by 1–2.5% due to air pollution [73].

### Recommendations

A national policy across the three countries is recommended to focus on numerous action plans as listed. These include but are not limited to establishing a consensus between state- and nation-wide bodies on transportation and transboundary pollution [74]. Stringent emissions and air-control policies in accordance with institutional frameworks and with regulatory compliance are necessary. Furthermore, the design and standards of vehicles ought to focus on low emissions and/or hybrid types. The public and private sectors ought to establish working groups to account for industrial power-based and automobile emissions. The stakeholder groups must link with the public health sectors and agricultural committees.

A replication of the self-monitoring and reporting (SMART) program as seen in the Pakistani industrial sector in March 2006, maybe reverse-engineered in other South Asian countries [75]. Technical assistance protocols may aid in mitigating cross-boundary multi-factorial effects of air pollution. The South Asian Association for Regional Cooperation (SAARC) forum may be a possible central system for accounting for and controlling air pollution reduction and control in the region [76]. With imminent impacts on public health, population viability, and personal well-being, a call to action must be made to strengthen the framework of air pollution and reduction in South Asia with legally binding instruments [77,78,79]. This will allow and encourage national and regional governments to promote clean emissions and adequate allocation of finances with country budgets allocated to the purpose. The objectives would be to protect the ecosystem and human health by reducing air pollution (i.e., nitrogen oxides) targets [80].

## 5. Conclusions

This study monitored climate indicators and changes between 1995 and 2021 with estimates given until 2100. In summary, this study had four key areas of interest. The first included the power sector’s use of fossil and coal in addition to carbon dioxide emissions. The second was to geographically map areas with high nitrogen dioxide tropospheric column along with reporting of AQI. The third was electricity generation and power sectors trends in the region compared to the world. The fourth was a projection of weather changes including the heat index and the annual SPEI drought index between the years 2020 and 2100.

We find that there is an urgent need to monitor and regulate power emissions, transportation usage, urbanization actions, and the adoption of renewable electricity generation in the region. While South Asia has been undergoing hastened economic and social development, the air quality and living standards of residents in the region has been decreasing due to heat waves, unexpected floodings, or in certain cases extreme rain. The symbiosis of living beings in the region, along with sources of food security (i.e., food security), are under threat. Not only international but national agencies have become aware of the contributors to climate change, but it is only with concerted actions in the SAARC region that progress may be made in the foreseeable future.

## Figures and Tables

**Figure 1 ijerph-19-17039-f001:**
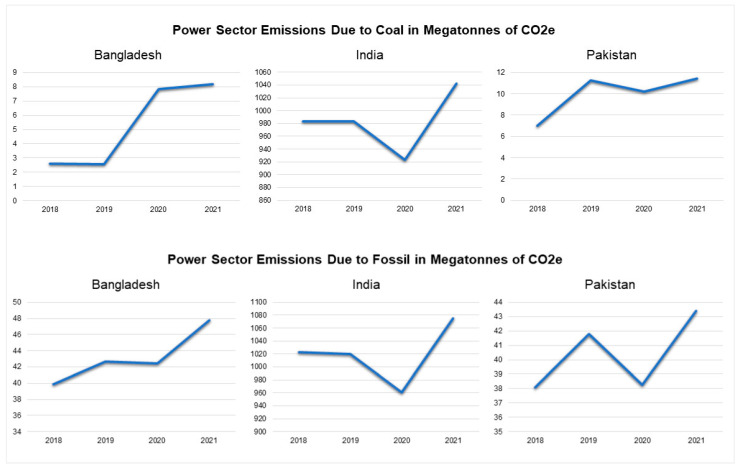
Power sector emissions as a consequence of coal and fossil use in megatons of CO_2_ (2018–2021) [43].

**Figure 2 ijerph-19-17039-f002:**
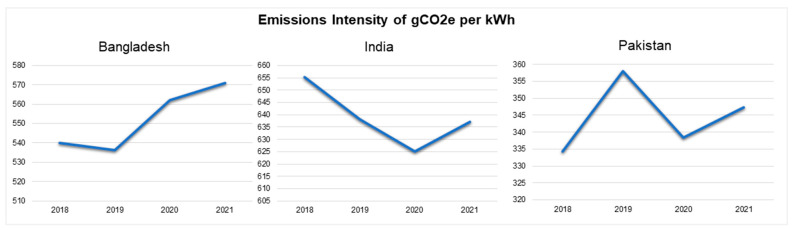
Emissions intensity of CO_2_ per kWh (2018–2021) [43].

**Figure 3 ijerph-19-17039-f003:**
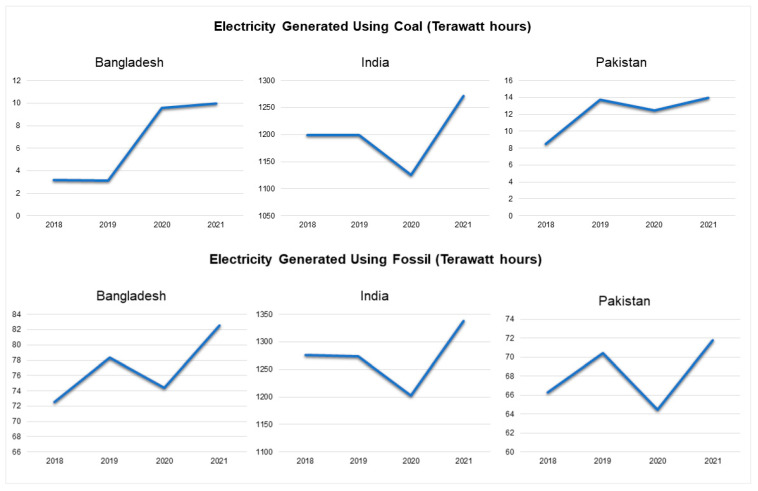
Electricity generated using coal and fossil in Terawatt hours (2018–2021) [43].

**Figure 4 ijerph-19-17039-f004:**
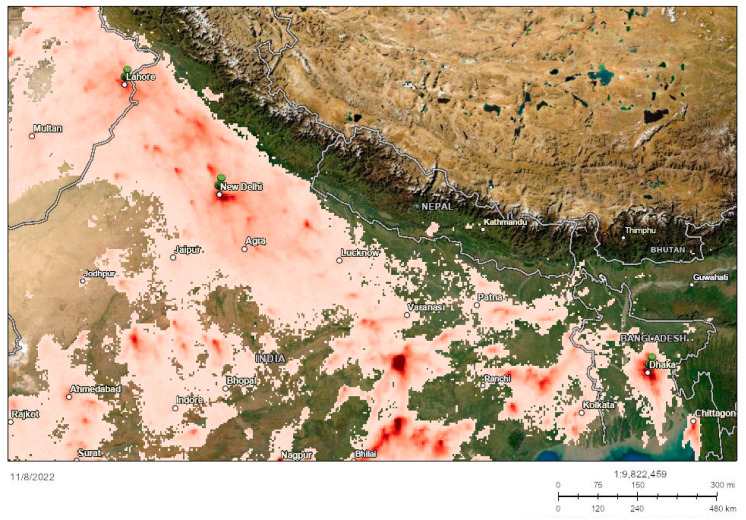
GIS mapping of low air quality geographic ‘spots’ in the South Asian region. The GIS image was generated on 11/8/2022, with data sources comprising the European Commission, European Space Agency, Esri, and the UKRI GCRF South Asian Nitrogen Hub [44,45].

**Figure 5 ijerph-19-17039-f005:**
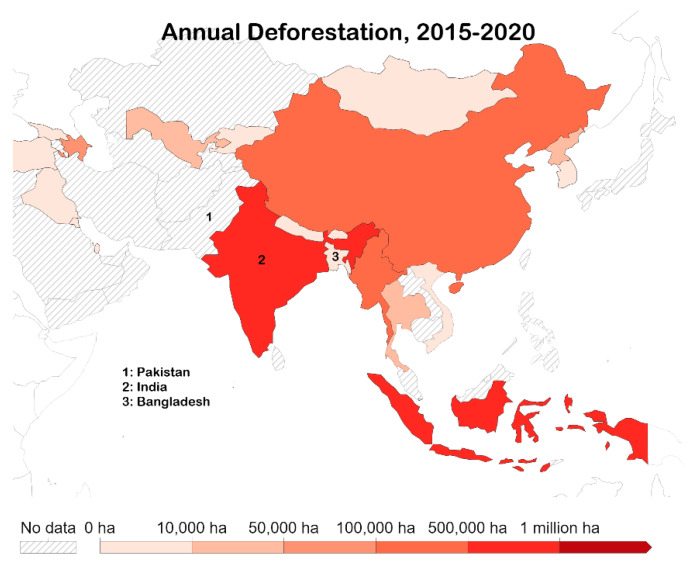
Annual deforestation between 2015–2020 in the South Asian region [46].

**Figure 6 ijerph-19-17039-f006:**
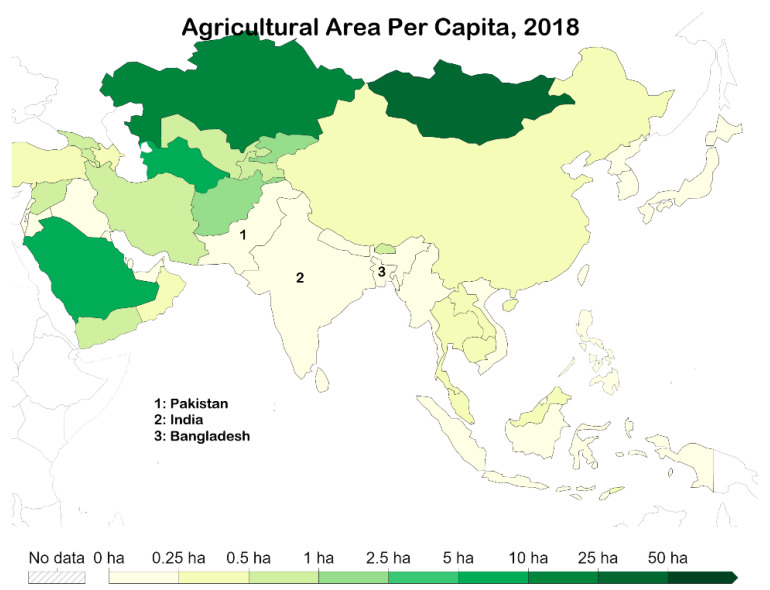
Agricultural area per capita—2018, in South Asia [46].

**Figure 7 ijerph-19-17039-f007:**
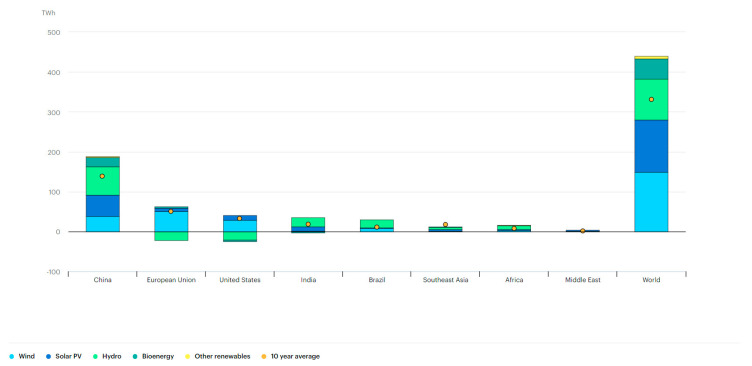
Annual growth of renewable electricity generation by region and technology, 2019 [47].

**Figure 8 ijerph-19-17039-f008:**
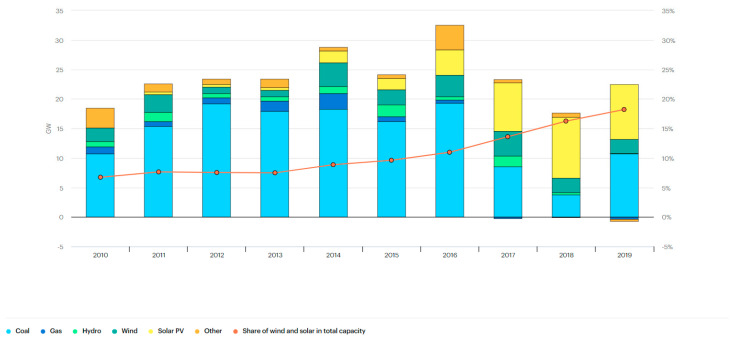
Annual power sector capacity additions in India, 2010–2019 [48].

**Figure 9 ijerph-19-17039-f009:**
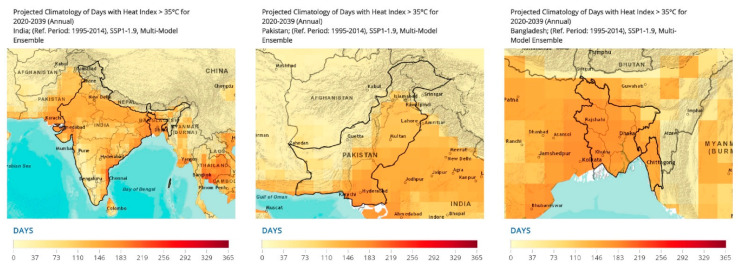
Projected Climatology of Days with Heat Index >35 °C for 2020–2039 (Annual) [49]. Annual projections for the years 2020–2039 are made for the heat index (>35 °C). A reference period of 1995–2014 was utilized. The SSP1–1.9, multi-model ensemble was utilized. The projections are scaled for 365 days as depicted in the figure.

**Figure 10 ijerph-19-17039-f010:**
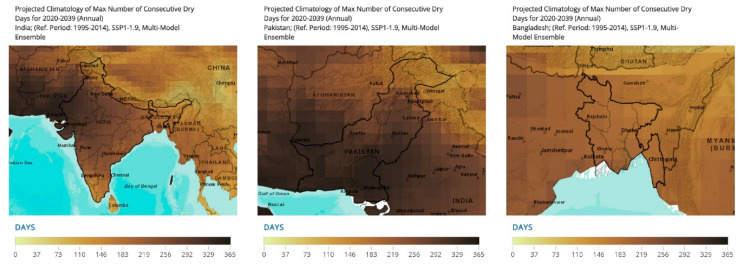
Projected climatology of the maximum number of consecutive dry days for 2020–2039 [49]. Annual projections for the years 2020–2039 are made for the maximum number of consecutive dry days. A reference period of 1995–2014 was utilized. The SSP1-1.9, multi-model ensemble was utilized. The projections are scaled for 365 days as depicted in the figure.

**Figure 11 ijerph-19-17039-f011:**
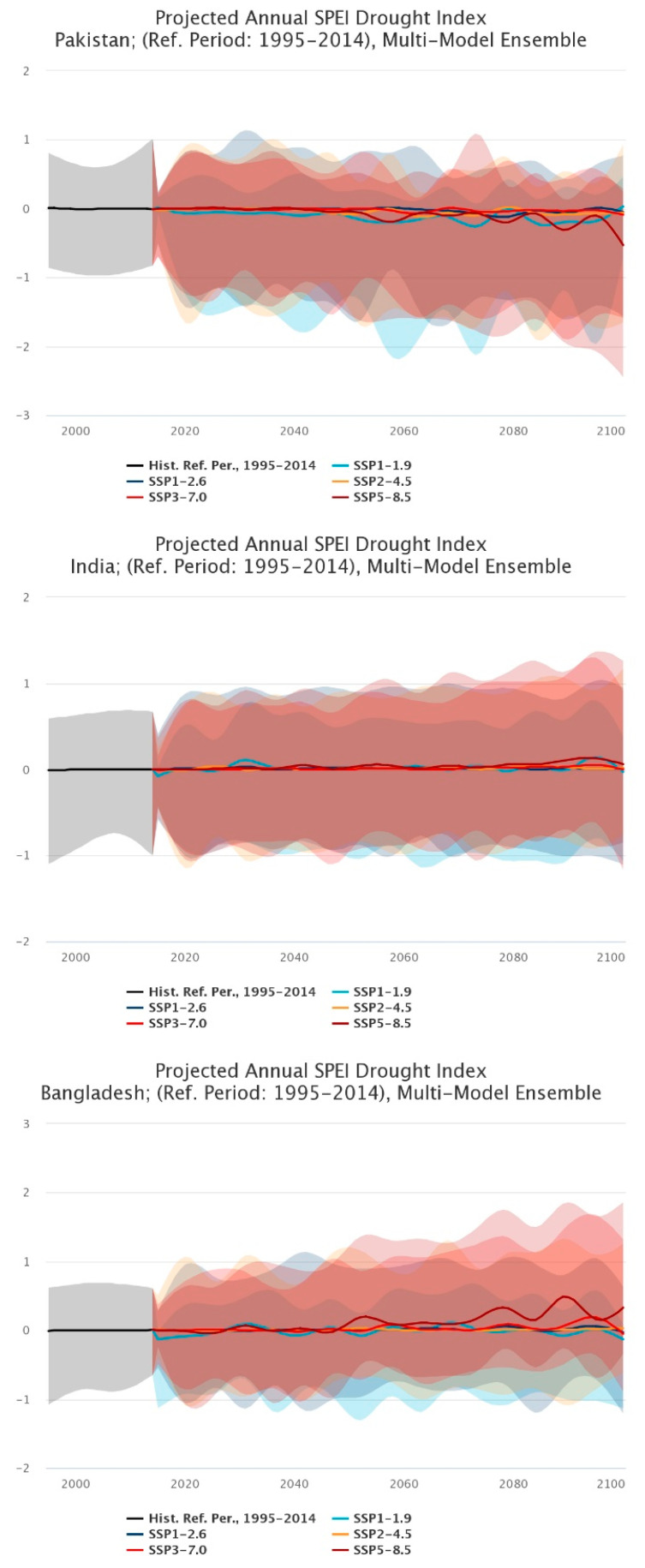
Projected Annual SPEI Drought Index (Reference Period: 1995–2014) sequential analysis trends for the 21st century [49].

**Figure 12 ijerph-19-17039-f012:**
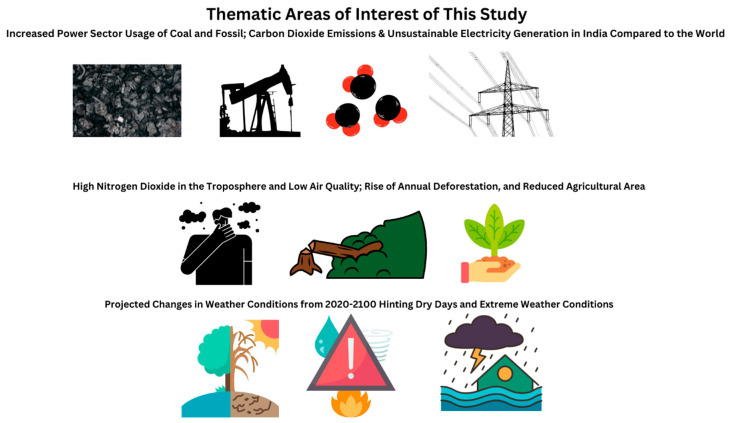
Thematic areas of interest of this study.

## Data Availability

All data utilized for the purpose of this study are available publicly and online in the enlisted databases.

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
