# Peer review of "Indicators of Climate Change, Geospatial and Analytical Mapping of Trends in India, Pakistan and Bangladesh: An Observational Study"

_ijerph, 2022, doi:10.3390/ijerph192417039_

Round 1

Reviewer 1 Report

This manuscript presents an interesting study of “Indicators of Climate Change, Geospatial and Analytical Mapping of Trends in India, Pakistan and Bangladesh: An Observational Study”.

Overall, it combines a nice array of data sources to present how the power sectors and carbon dioxide emissions, maps nitrogen dioxide density across three countries (Pakistan, India, and Bangladesh) are linked. This it helps in understanding electricity generation trends and projects weather changes through 2100. The data collected for this study covering the time period (1995 to 2021) from the databases of the International Energy Agency, the World Bank, the UN Food and Agricultural Organization.

The paper clearly presents its objectives and can make an important contribution to planners. Furthermore, the materials and methods are detailed and allow such procedures to be applied in other areas of interest. Similarly, the results are detailed and consistent with the procedures used. However, there are some aspects of the manuscript that I believe should be addressed by the authors before publication. Firstly, the text needs extensive editing to correct typographical, grammatical and spelling mistakes. Examples are too many to be included in this report, but I could select a few just from the manuscript:

1.     Please correct this sentence in the abstract, three countries are mentioned in this sentence, “The results present increased usage of coal and fossil fuels across Pakistan, India, and Bangladesh which are both large contributors to carbon dioxide emissions.

2.     Please rephrase this sentence, “Between 2020 and 2100, there are expected to be increased days with heat index >35°C, in addition to consecutive dry days; this is also compounded with increased flooding in certain regions of the three countries.

3.     The key-words should be alphabetically arranged.

4.     Line #53-56, Is this sentence mean ozone level increase in the troposphere or stratosphere? Please correct this sentence, “The WMO report also outlines that the ozone level increase is a direct consequence of emissions from fossil fuel combustion, however, 20% of the change is due to climate change, which is caused by deforestation combined with agriculture and other land use changes and energy-related carbon-dioxide emissions from buildings.

5.     Line # 59, Please reference this sentences, “South Asia is one of the most vulnerable regions to climate shocks. I suggest adding the reference here (Sultan et al. 2022b).

·       Sultan H, Zhan J, Rashid W, Chu X, Bohnett E. 2022. Systematic Review of Multi-Dimensional Vulnerabilities in the Himalayas. International Journal of Environmental Research and Public Health 19:12177. https://doi.org/10.3390/ijerph191912177.

6.     Pakistan and India have same geographical location and are located along the tectonic plate boundary. Please correct this sentence, “Pakistan rests on a tectonic plate boundary meaning that earthquakes are common occurrences”.

7.     Line # 66-67, please correct this sentence, “India is the second largest consumer of coal worldwide which is centrally used for electricity production and compounds air pollution [30,31].

8.     Line # 72-73, this sentence need to be referenced and the following reference may be added here, “South Asia has large variability in geographical features, climate and landscapes”(Sultan et al. 2022a).

        Sultan, H.; Rashid, W.; Shi, J.; Rahim, I. u.; Nafees, M.; Bohnett, E.; Rashid, S.; Khan, M. T.; Shah, I. A.; Han, H.; Ariza-Montes, A., Horizon Scan of Transboundary Concerns Impacting Snow Leopard Landscapes in Asia. Land 2022, 11, (2), 248-269. https://doi.org/10.3390/land11020248

9.     Line # 77-81, Please correct this sentence , “In this observational study, the aim is to understand the linkage between power sectors in the region and carbon dioxide emissions, map areas with nitrogen dioxide tropospheric columns and correlated air quality indices, understand the power sector and all energy generation trends in India and South Asia, and lastly to project weather changes throughout the 21st century” and may be corrected as, “The aim of this observational study is to understand the linkage between power sectors in the region (India, Pakistan, and Bangladesh) and carbon dioxide emissions, map areas with nitrogen dioxide tropospheric columns and correlated air quality indices, understand the power sector and all energy generation trends in and lastly to project weather changes throughout the 21st century”.

10.  Line # 85-87, Please check and correct the spelling “data backs”.

11.  Line # 93-94, Please check and correct the spellings “country-specific draught

12.  Line # 131, Please correct the “CO2” as CO2

13.  Please use a uniform terminology throughout the manuscript. The carbon dioxide is used in three different forms e.g, “ CO2” , CO2 or carbon dioxide throughout the manuscript.

14.  Please revise and correct this sentence, “Bangladesh has upgraded from the developing nation category to the least developed country category; as an emerging economy.

15.  Figure 9 and Figure 10, Please add legends to these map.

16.  Line # 317, Please correct this sentence, “India is one of the most populous countries on earth”. As India is the second populous country.

17.  Line 333-335, Please correct this sentence, “Historical also estimates that the electricity consumption per capita will increase by 22 times by 2050 as compared to 2014 [51].

18.  Please correct the “NO2” as “NO2

19.  Line 358-362, the reference [62] is similar for the three sentences, please add this reference at the end.

20.  Please check abbreviations such as SAARC, NOX etc that should be written in full at first appearance, then followed by abbreviated form throughout the manuscript. However, currently they are used interchangeably in full and abbreviations at different locations of the manuscript.

Author Response

Note to Reviewer: I thank you for your valuable time and input into reviewing our paper and providing essential comments. This has drastically improved our paper, and your time and dedication is valuable to us.

Regards,

Dr Zouina Sarfraz

---

Reviewer 1 Comments and Author Responses:

This manuscript presents an interesting study of “Indicators of Climate Change, Geospatial and Analytical Mapping of Trends in India, Pakistan and Bangladesh: An Observational Study”.

Overall, it combines a nice array of data sources to present how the power sectors and carbon dioxide emissions, maps nitrogen dioxide density across three countries (Pakistan, India, and Bangladesh) are linked. This it helps in understanding electricity generation trends and projects weather changes through 2100. The data collected for this study covering the time period (1995 to 2021) from the databases of the International Energy Agency, the World Bank, the UN Food and Agricultural Organization.

The paper clearly presents its objectives and can make an important contribution to planners. Furthermore, the materials and methods are detailed and allow such procedures to be applied in other areas of interest. Similarly, the results are detailed and consistent with the procedures used. However, there are some aspects of the manuscript that I believe should be addressed by the authors before publication. Firstly, the text needs extensive editing to correct typographical, grammatical and spelling mistakes. Examples are too many to be included in this report, but I could select a few just from the manuscript:

Comment 1.     Please correct this sentence in the abstract, three countries are mentioned in this sentence, “The results present increased usage of coal and fossil fuels across Pakistan, India, and Bangladesh which are both large contributors to carbon dioxide emissions.

Author Response: Thank you for your helpful comment. The required correction has been made. I invite you to have a look.

Comment 2.     Please rephrase this sentence, “Between 2020 and 2100, there are expected to be increased days with heat index >35°C, in addition to consecutive dry days; this is also compounded with increased flooding in certain regions of the three countries.

Author Response: Thank you for your helpful comment. The required correction has been made. I invite you to have a look.

Comment 3.     The key-words should be alphabetically arranged.

Author Response: Thank you for your helpful comment. The required correction has been made. I invite you to have a look.

Comment 4.     Line #53-56, Is this sentence mean ozone level increase in the troposphere or stratosphere? Please correct this sentence, “The WMO report also outlines that the ozone level increase is a direct consequence of emissions from fossil fuel combustion, however, 20% of the change is due to climate change, which is caused by deforestation combined with agriculture and other land use changes and energy-related carbon-dioxide emissions from buildings.

Author Response: Thank you for your helpful comment. The required correction has been made. I invite you to have a look. I mean the troposphere. 

Comment 5.     Line # 59, Please reference this sentences, “South Asia is one of the most vulnerable regions to climate shocks. I suggest adding the reference here (Sultan et al. 2022b).

  •       Sultan H, Zhan J, Rashid W, Chu X, Bohnett E. 2022. Systematic Review of Multi-Dimensional Vulnerabilities in the Himalayas. International Journal of Environmental Research and Public Health 19:12177. https://doi.org/10.3390/ijerph191912177.

Author Response: Thank you for your helpful comment. The required correction has been made. I invite you to have a look.

Comment 6.     Pakistan and India have same geographical location and are located along the tectonic plate boundary. Please correct this sentence, “Pakistan rests on a tectonic plate boundary meaning that earthquakes are common occurrences”.

Author Response: Thank you for your helpful comment. The required correction has been made. I invite you to have a look.

Comment 7.     Line # 66-67, please correct this sentence, “India is the second largest consumer of coal worldwide which is centrally used for electricity production and compounds air pollution [30,31].

Author Response: Thank you for your helpful comment. The required correction has been made. I invite you to have a look.

Comment 8.     Line # 72-73, this sentence need to be referenced and the following reference may be added here, “South Asia has large variability in geographical features, climate and landscapes”(Sultan et al. 2022a).

  •       Sultan, H.; Rashid, W.; Shi, J.; Rahim, I. u.; Nafees, M.; Bohnett, E.; Rashid, S.; Khan, M. T.; Shah, I. A.; Han, H.; Ariza-Montes, A., Horizon Scan of Transboundary Concerns Impacting Snow Leopard Landscapes in Asia. Land 2022, 11, (2), 248-269. https://doi.org/10.3390/land11020248

Author Response: Thank you for your helpful comment. The required citation and correction has been made. I invite you to have a look.

Comment 9.     Line # 77-81, Please correct this sentence , “In this observational study, the aim is to understand the linkage between power sectors in the region and carbon dioxide emissions, map areas with nitrogen dioxide tropospheric columns and correlated air quality indices, understand the power sector and all energy generation trends in India and South Asia, and lastly to project weather changes throughout the 21st century” and may be corrected as, “The aim of this observational study is to understand the linkage between power sectors in the region (India, Pakistan, and Bangladesh) and carbon dioxide emissions, map areas with nitrogen dioxide tropospheric columns and correlated air quality indices, understand the power sector and all energy generation trends in and lastly to project weather changes throughout the 21st century”.

Author Response: Thank you for your helpful comment. The required correction has been made. I invite you to have a look.

Comment 10.  Line # 85-87, Please check and correct the spelling “data backs”.

Author Response: Thank you for your helpful comment. The required correction has been made. I invite you to have a look.

Comment 11.  Line # 93-94, Please check and correct the spellings “country-specific draught

Author Response: Thank you for your helpful comment. The required correction has been made. I invite you to have a look.

Comment 12.  Line # 131, Please correct the “CO2” as CO2

Author Response: Thank you for your helpful comment. The required correction has been made throughout the manuscript. I invite you to have a look.

Comment 13.  Please use a uniform terminology throughout the manuscript. The carbon dioxide is used in three different forms e.g, “ CO2” , CO2 or carbon dioxide throughout the manuscript.

Author Response: Thank you for your helpful comment. The required correction has been made throughout the manuscript. I invite you to have a look.

Comment 14.  Please revise and correct this sentence, “Bangladesh has upgraded from the developing nation category to the least developed country category; as an emerging economy.

Author Response: Thank you for your helpful comment. The required correction has been made. I invite you to have a look.

  1. Figure 9 and Figure 10, Please add legends to these map.

Author Response: Thank you for your helpful comment. Legends have been added to both maps.

Comment 16.  Line # 317, Please correct this sentence, “India is one of the most populous countries on earth”. As India is the second populous country.

Author Response: Thank you for your helpful comment. The required correction has been made. I invite you to have a look.

Comment 17.  Line 333-335, Please correct this sentence, “Historical also estimates that the electricity consumption per capita will increase by 22 times by 2050 as compared to 2014 [51].

Author Response: Thank you for your helpful comment. The required correction has been made. I invite you to have a look.

Comment 18.  Please correct the “NO2” as “NO2

Author Response: Thank you for your helpful comment. The required correction has been made throughout the manuscript. I invite you to have a look.

Comment 19.  Line 358-362, the reference [62] is similar for the three sentences, please add this reference at the end.

Author Response: Thank you for your helpful comment. The required correction has been made. I invite you to have a look.

  1. Please check abbreviations such as SAARC, NOX etc that should be written in full at first appearance, then followed by abbreviated form throughout the manuscript. However, currently they are used interchangeably in full and abbreviations at different locations of the manuscript.

Author Response: Thank you for your helpful comment. All discrepancies have been reviewed multiple times and are currently fixed.

Reviewer 2 Report

A well written paper with extensive research. Check spell before publication as some expressions are confusing, e.g., line 333, 'Historical also estimates...'

Author Response

Note to Reviewer: I thank you for your valuable time and input into reviewing our paper and providing essential comments. This has drastically improved our paper, and your time and dedication is valuable to us.

Regards,

Dr Zouina Sarfraz

Reviewer 2 Comments and Author Responses:

A well written paper with extensive research. Check spell before publication as some expressions are confusing, e.g., line 333, 'Historical also estimates...'

Author Response: Thank you for your helpful comment. The required corrections for expression have been made. I invite you to have a look at the revised manuscript.

Reviewer 3 Report

Comments:

This manuscript quantitatively assesses climate risk in South Asia (mainly India, Pakistan, and Bangladesh) based on multiple geological hazard databases and predicts weather changes in the 21st century. The emphasis is on the indicators of climate change that ascertained both causation and effect. It is of great significance to summarize climate datasets of south Asia to prevent disaster in a climate fast-changing context. However, the method described in this paper is not clear and complete, the content and analysis lack depth, and there are many mistakes in the presentation. Therefore, I recommend major revision. Following are the detailed comments:

1)        In Abstract, line 23, authors are suggested to point out methods they used, not only give their results.

2)        Descriptions in MS should be consistent, for example, in line 23, it is “with Nitrogen dioxide”, while in other places, it becomes “nitrogen dioxide”. There’s no need to be capitalized “nitrogen”.

3)        In line 24, “with heat index…” should be “with a heat index…”.

4)        In sentences in lines 18 to 20 and lines 26 to 28, please be especially careful with your expressions if a sentence exists more than two predicates. The expressions in these two places may have problems that the meaning is not clearly expressed. Maybe this part can be improved.

5)        In Introduction, too many background statements. The research results of predecessors should be given.

6)        In line 141, there is a typesetting error here. The indentation is inconsistent with other subheadings of the same level.

7)        All of the figures in MS should be mentioned in the article.

8)        In line 179, “With the geospatial map generated on 8th November 2022, densely polluted air localities were mapped in South Asia.” Where does this map come from? References should be noted here.

9)        In 191 “Since 2010, the net forest loss worldwide was 4.7 million hectares per year, however, the deforestation rates were higher” Where does this result come from? References should be noted here. Similar issues of missing references are found in the results section, such as in lines 143, 159, 205, and so on.

10)     In Figure 5, the text is too small, and the title of the figure is not clearly described.

11)     In Figures 9 and 10, it can be seen that these two figures are of the same series, but the color scale is not clear. Figure 10 also has data from the non-study area.

12)     In lines 294 to 302, authors are suggested to introduce their work in results or the final part of the introduction, not in discussion. Maybe this part can be improved!

13)     In Materials and Methods, key methods used in MS need to be described in detail to some extent rather than simply directed to data reports. In general, this part is vague.

14)     In Conclusion, the conclusion is general like the background introduction. Please be forward-looking and give key results.

15)     From the last paragraph of the introduction, the first paragraph of the discussion, and the conclusion, it can be seen that the author wants to describe the key climate indicators in South Asia, but too many aspects were investigated with no main idea. Also, the authors do not have their own main data analysis methods. This manuscript is more like a review, but it needs to be organized to combine the various parts.

Author Response

Note to Reviewer: I thank you for your valuable time and input into reviewing our paper and providing essential comments. This has drastically improved our paper, and your time and dedication is valuable to us.

Regards,

Dr Zouina Sarfraz

Reviewer 3 Comments and Author Responses:

This manuscript quantitatively assesses climate risk in South Asia (mainly India, Pakistan, and Bangladesh) based on multiple geological hazard databases and predicts weather changes in the 21st century. The emphasis is on the indicators of climate change that ascertained both causation and effect. It is of great significance to summarize climate datasets of south Asia to prevent disaster in a climate fast-changing context. However, the method described in this paper is not clear and complete, the content and analysis lack depth, and there are many mistakes in the presentation. Therefore, I recommend major revision. Following are the detailed comments:

Comment 1. In Abstract, line 23, authors are suggested to point out methods they used, not only give their results.

Author response: Thank you for your important comment. The required addition has been made.

Raw data was obtained for climate indicators, which were entered into Microsoft Excel. Geospatial trends were generated in the ArcGIS geostatistical tool by adopting the ordinary kriging method to interpolate and create continuous surfaces depicting concentration of nitrogen dioxide in the three countries.

Comment 2. Descriptions in MS should be consistent, for example, in line 23, it is “with Nitrogen dioxide”, while in other places, it becomes “nitrogen dioxide”. There’s no need to be capitalized “nitrogen”.

Author response: Thank you for your important comment. The discrepancies have been resolved throughout the manuscript.

Comment 3. In line 24, “with heat index…” should be “with a heat index…”.

Author response: Thank you for your important comment. The required correction has been made.

Comment 4. In sentences in lines 18 to 20 and lines 26 to 28, please be especially careful with your expressions if a sentence exists more than two predicates. The expressions in these two places may have problems that the meaning is not clearly expressed. Maybe this part can be improved.

Author response: Thank you for your important comment. The required correction has been made.

Comment 5. In Introduction, too many background statements. The research results of predecessors should be given.

Fresh information based on research results of predecessors has been added:

“Current literature cites reports of geospatial and analytical studies conducted in the region. Syed and colleagues (2021) assessed changes in season trends and spatial patterns of temperature indicators between 1979 and 2014 [33]. The authors used datasets by Climate Forecast System Reanalysis (CFSR). The authors found significant trends of autumn and spring for minimum temperature in Punjab, Pakistan [33]. Overall, the authors reported the reasons of sustainability imbalance in Punjab, Pakistan. These included fluctuations in humidity and trends of high temperature [33]. Javid et al. (2019) used the remote sensing climate compound index (RSCCI) to measure humidity, aridity, and semi-aridity in Pakistan [34]. The authors found an increase in humid regions and wetlands in Pakistan, 9.7% and 1.9% respectively [34]. There was a reduction in intense drought (4.2%) in the country between 2000 and 2018 [34]. The study concluded with predicting that the next 30 years may lead to dramatic climatic changes in Pakistan [34]. Ali et al. (2021) evaluated spatio-temporal rainfall patterns and trends in Pakistan between 1961 and 2020 by including 82 rainfall stations [35]. The authors use the Bayesian kriging regression prediction (EBKRP) technique [35]. The lowest rainfalls were prediction in Baluchistan. There was a significant downward trend of temporal rainfall evaluation in Sindh [35]. The study sheds light on environmental planning in the region [35].”

Comment 6. In line 141, there is a typesetting error here. The indentation is inconsistent with other subheadings of the same level.

Author response: Thank you for your important comment. The required update to the typesetting has been made.

Comment 7. All of the figures in MS should be mentioned in the article.

Author response: Thank you for your important comment. The required mentioning of all figures has been made throughout the manuscript.

Comment 8. In line 179, “With the geospatial map generated on 8th November 2022, densely polluted air localities were mapped in South Asia.” Where does this map come from? References should be noted here.

Author response: Thank you for your important comment. The required referencing has been made.

“The GIS image was generated on 11/8/2022, with data sources comprising the European Commis-sion, European Space Agency, Esri, and the UKRI GCRF South Asian Nitrogen Hub, [41,42].”

Comment 9. In 191 “Since 2010, the net forest loss worldwide was 4.7 million hectares per year, however, the deforestation rates were higher” Where does this result come from? References should be noted here. Similar issues of missing references are found in the results section, such as in lines 143, 159, 205, and so on.

Author response: Thank you for your important comment. The required mentioning of citations has been made throughout these sections. I invite you to review it.

Comment 10. In Figure 5, the text is too small, and the title of the figure is not clearly described.

Author response: Thank you for your important comment. The figure has been regenerated with clear titles and labels.

Comment 11. In Figures 9 and 10, it can be seen that these two figures are of the same series, but the color scale is not clear. Figure 10 also has data from the non-study area.

Author response: Thank you for your important comment. The figure has been regenerated with clear color scales. The image type is naturally pixelated as it is showing us the differences in the included regions for days. It is not being generated without the pixelated image. However, I have increased the quality and it is broken down to India, Pakistan, and Bangladesh (3 countries). The non-study area cannot be erased in the software and serves as a good control for the viewer to my knowledge.

Comment 12. In lines 294 to 302, authors are suggested to introduce their work in results or the final part of the introduction, not in discussion. Maybe this part can be improved!

Author response: Thank you for your important comment. The required change has been made. I have moved the paragraph to the beginning of the results section.

Comment 13. In Materials and Methods, key methods used in MS need to be described in detail to some extent rather than simply directed to data reports. In general, this part is vague.

Author response: The methods section has been updated with the following:

“Geographic Information System (GIS) tool was used to generate spatially distributed air pollution maps in India, Pakistan, and Bangladesh. In order to map the pollution levels using nitrogen dioxide, the ArcGIS geostatistical analysis tool was used due to its ability in handling wide data formats and layers in digital maps using frameworks of spatio-temporal analysis. The ordinary kriging method was adopted to interpolate and create continuous surfaces from the database of measured concentration across the locations [41,42]. Further analysis was conducted to ascertain the usage of different energy types in the power sector and reliance on fossil and coal as modes of produc-tion. All data was entered into Microsoft Excel and output plots were generated to depict the sequential trends in the various indicators.”

Thank you for your suggestion.

Comment 14. In Conclusion, the conclusion is general like the background introduction. Please be forward-looking and give key results.

Author response: I agree. I have summarized the findings of our study and have updated it in the conclusion: “In summary, this study had four key areas of interest. The first included the power sector’s use of fossil and coal in addition to carbon dioxide emissions. The second was to geographically map areas with high nitrogen dioxide tropospheric column along with reporting of AQI. The third was electricity generation and power sectors trends in the region compared to the world. The fourth was a projection of weather changes including the heat index and the annual SPEI drought index between the years 2020 and 2100.

Comment 15. From the last paragraph of the introduction, the first paragraph of the discussion, and the conclusion, it can be seen that the author wants to describe the key climate indicators in South Asia, but too many aspects were investigated with no main idea. Also, the authors do not have their own main data analysis methods. This manuscript is more like a review, but it needs to be organized to combine the various parts.

Author response: I agree with all your prior comments, however, the current comment must be reassessed. I have received excellent feedback by the reviewer panel. However, I have made updates to the paper throughout. Please have a look.

Reviewer 4 Report

This work collects diverse environmental information, geolocated in India, Pakistan and Bangladesh. The objective is to generate a set of indicators that allow an analysis of the projected impact of climate change in that area.

The work is well written, but the purpose of the study needs to be more clearly defined. 

In the Materials section, it is necessary to define the tool used to manage the GIS. In addition, it would be desirable to include a graph showing the interrelationships between the various indicators.

The set of results obtained suggest that both the discussion and the conclusions should be expanded.

Author Response

Note to Reviewer: I thank you for your valuable time and input into reviewing our paper and providing essential comments. This has drastically improved our paper, and your time and dedication is valuable to us.

Regards,

Dr Zouina Sarfraz

Reviewer 4 Comments and Author Responses: 

This work collects diverse environmental information, geolocated in India, Pakistan and Bangladesh. The objective is to generate a set of indicators that allow an analysis of the projected impact of climate change in that area.

Comment 1: The work is well written, but the purpose of the study needs to be more clearly defined. ]

Author Response: It has been clarified in the introduction.

“The motivation behind this study is to prioritize climate adaptation and subse-quently adapt strategies. There is an urgent need for quantitative-regional based as-sessment of climate risks. The aim of this observational study is to understand the linkage between power sectors in the region (India, Pakistan, and Bangladesh) and carbon dioxide emissions, map areas with nitrogen dioxide tropospheric columns and correlated air quality indices, understand the power sector and all energy generation trends in and lastly to project weather changes throughout the 21st century.”

Comment 2: In the Materials section, it is necessary to define the tool used to manage the GIS. In addition, it would be desirable to include a graph showing the interrelationships between the various indicators.

Author response: The tool used to manage GIS has been mentioned in the methods: Geographic Information System (GIS) tool was used to generate spatially distributed air pollution maps in India, Pakistan, and Bangladesh. In order to map the pollution levels using nitrogen dioxide, the ArcGIS geostatistical analysis tool was used due to its ability in handling wide data formats and layers in digital maps using frameworks of spatio-temporal analysis. The ordinary kriging method was adopted to interpolate and create continuous surfaces from the database of measured concentration across the locations [41,42]. 

Comment 3: The set of results obtained suggest that both the discussion and the conclusions should be expanded.

Author response: I have expanded the conclusions section and have clarified portions of the discussion for easy viewing (1200+ words at present).

Round 2

Reviewer 3 Report

Thanks for authors' revision. I have no further comments.